# Effect of Pre-Storage CO_2_ Treatment and Modified Atmosphere Packaging on Sweet Pepper Chilling Injury

**DOI:** 10.3390/plants12030671

**Published:** 2023-02-03

**Authors:** Abiodun Samuel Afolabi, In-Lee Choi, Joo Hwan Lee, Yong Beom Kwon, Hyuk Sung Yoon, Ho-Min Kang

**Affiliations:** 1Interdisciplinary Program in Smart Agriculture, Kangwon National University, Chuncheon 24341, Republic of Korea; 2Agricultural and Life Science Research Institute, Kangwon National University, Chuncheon 24341, Republic of Korea; 3Waksman Institute of Microbiology, Rutgers the State University of New Jersey, Piscataway, NJ 08854, USA

**Keywords:** calyx browning, electrolyte leakage, ethylene production rate, malondialdehyde, respiration rate

## Abstract

The effect of 10% CO_2_ pre-storage treatment for 12, 24, and 48 h alongside modified atmosphere packaging (MAP) on chilling injury was determined in this study. This study found significant interactions between chilling injuries and cell membrane damage indicators. The results show that chilling injuries can be somewhat reduced by the use of CO_2_ treatment for sweet peppers. It was noticed that the fruit’s respiration rate increased as the treatment duration increased immediately after the treatments, while the resultant did not affect the ethylene production rate, electrolyte leakage, or malondialdehyde. Similarly, after cold storage and on the final day, no really significant differences were shown in all those parameters except for the weight loss rate, chilling injury, calyx browning, and firmness, which were at the poorest state in the control group. Of all the treatments in this study, MAP appeared to be the best treatment, and preference may be given to the 24 h treatment of pretreated fruits. Weight loss, firmness, calyx browning, and chilling injury were maintained best in MAP due to the presence of CO_2_ and high humidity.

## 1. Introduction

One of the many tropical and subtropical horticultural products with a low tolerance to chilling injury is the sweet pepper (*Capsicum annum*). Injury is more common at temperatures lower than 7 °C, with typical symptoms including surface pitting, calyx browning, seed browning, and shriveling caused by water loss [1]. Babellahi [2] estimated that this accounts for 25–35% of total production losses. Extreme measures have been taken over the past two decades to resist this, and a number of physical and chemical treatments have succeeded. Modified atmosphere packaging (MAP) is one of them [3]. In this method, the presence of more CO_2_ and the reduction of O_2_ were found to protect the fruits from damage. However, there has been a lot of contentious conjecture made about peppers in this condition. For instance, a study attributed a reduced chilling injury to the CO_2_ content in MAP conditions [3]. Another study determined that sweet peppers have a maximum tolerance level of 5% CO_2_ in this condition [4,5], despite another study claiming that up to 20% CO_2_ could reduce chilling injury rates. Meanwhile, Mathooko [6] showed that a high concentration of 60% CO_2_ in cucumbers causes other types of stress that increase the ethylene production rate and putrescine level, which conversely accumulate in fruits and are attributed to chilling injuries. Therefore, the inability to regulate CO_2_ content within the MAP may be something to worry about. Another factor impeding commercial acceptance is the decay and off-flavor development associated with some of the films [7].

The use of high CO_2_ treatment for a short time has gained attention in postharvest management as well. This is due to its ability to reduce firmness loss and retard senescence by reducing respiration and ethylene production rates [8]. It does this primarily by inhibiting the ethylene action competitively; this is because it is related to ethylene metabolism and inhibits its oxidation [9]. In other words, it slows down respiratory movement by inhibiting the succinic oxidase complex and succinic dehydrogenase, thereby slowing down aging [9]. Moreover, because of its effect on redox and the cellular membrane, short-term anoxia has been shown to alleviate postharvest disorders [10]. A study shows it to be beneficial in that it increases resistance to bacterial and fungal attacks [11], which is most likely accomplished by inhibiting cell wall degradation. These, among many other reasons, have shown the cause of its general adoption.

Zucchini and persimmon, like peppers, show 10% CO_2_ tolerance in MAP conditions [12], and high concentrations of 40% and 90% CO_2_, respectively, have been found to reduce chilling injury in short periods of anoxic conditions [10,13]. Following that, short-term treatments should be tested for peppers. Peppers exposed to a variety of moderate stress conditions have shown less sensitivity to future stressors. Under CO_2_ stress, acetaldehyde reacts with soluble tannins to render them insoluble, increasing firmness [14]. This makes it possible for fruit to withstand additional stress. In our previous study, although unpublished, 10% CO_2_ was established as the best of various treatments, which is similarly supported by Wang [15] for CO_2_-treated sweet peppers. As a result, in this study, we investigated the best duration of 10% CO_2_ treatment to reduce chilling injury in fruits.

## 2. Results

### 2.1. Weight Loss and Firmness

A relationship has previously been shown to exist between weight loss and firmness [5]. Hence, the result is shown side by side in Figure 1. Aside from the MAP fruits that were difficult to monitor after unpacking on the 15th day, the highest weight loss rate was in the control group (untreated fruits) and the lowest was in the 24 h treatment of pretreated fruits (Figure 1A). This result is consistent with pretreated zucchini [3], while the primary reason for storing fruits in the MAP is to see how long-term CO_2_ affects fruit.

Before pretreatments, the fruit’s initial firmness was 43.47 ± 2.34 N. Firmness declined for all fruits on the last day but remained non-significant for the MAP fruits (Figure 1B). The 48-h-treated group comes next in the line of pretreated fruits, and the control group comes last. However, at *p* ˂ 0.05, no statistical significance was found for pretreated fruits.

### 2.2. Chilling Injury and Calyx Browning Index

Pre-storage 10% CO_2_ treatments and MAP were found to effectively reduce the chilling injury and the calyx browning index (Figure 2). They were seen to have positive correlations, and both followed an almost similar pattern, with the highest in the control group and the lowest in the MAP fruits (Figure 2A). Furthermore, a non-significant difference exists between the chilling injury shown in 48-h-treated fruit and MAP, which was the best, as well as in control and 12-h-treated fruit. Calyx browning was insignificant for pretreated fruits, irrespective of the duration of treatment, and was best and worst in the MAP and control, respectively (Figure 2B). Also, these differences weren’t highly significant on the 15th day but became noticeable at room temperature (Figure 3). 

### 2.3. Gas Production Rates and Concentration in Modified Atmosphere Packaging (MAP)

The fruit’s respiration and ethylene production rates are shown in Table 1 and Table 2. Immediately after the CO_2_ treatment, fruit respiration was noticed to increase based on the duration of the treatment, while the resultant effect did not show on the ethylene production rate. When fruit was moved to room temperature after 15 days, the 24-h-treated fruits had the lowest respiration rate, while the MAP fruits had the highest, and it was lowest in the 48-h-treated fruit on the 20th day. Throughout the room-temperature days, no significant difference was observed. Ethylene production rates varied similarly when fruits were moved to room-temperature storage. The 48 h treatment produced the least ethylene, followed by MAP, with the highest levels in the 12 h treatment. On the 20th day, it was highest in the control groups and the lowest in the 48 h treatment.

The O_2_ and CO_2_ content show a reversed pattern throughout the measuring time (Figure 4A,B). The 50,000 cc·m^−2^·day^−1^·atm^−1^ OTR film used kept the average CO_2_ content value at a maximum of 2.05 ± 0.51%. The ethylene content was at its peak on the 10th day, before declining prior to opening on the 15th day (Figure 4C).

### 2.4. Malondialdehyde (MDA) Assay and Electrolyte Leakage

At *p* ˂ 0.05, a non-significant difference in malondialdehyde (MDA) and electrolyte leakage was observed after CO_2_ treatment and on the final day of storage (Figure 5). The initial content of MDA was 0.38 ± 0.13 µM kg^−1^, and it ranged between 0.96 ± 0.08 and 0.83 ± 0.10 µM kg^−1^ for highest and lowest contents, respectively, on the last day (Figure 5A). Additionally, before treatments, the fruit electrolyte was 40.8 ± 7.56% and it increased for all fruits when rechecked after treatments, with no statistical difference between treatments (Figure 5B). It increased more on the last day, and it appeared lower in treated fruits with a longer duration of treatment. The highest ion leakage was in the control and 12 h treatments.

### 2.5. Soluble Solids Content

The soluble solids content was 7.43 ± 2.3° Brix before treatment, and Figure 6 depicts the content of soluble solids after CO_2_ treatment and on the final day. It increased when rechecked after CO_2_ treatment. It increased for all fruits but was at a lower rate for 24 h and 48 h treatments compared to others. Although it was not monitored during storage because no highly significant differences were seen in respiration or ethylene production rates (Table 1 and Table 2), it varied for all fruits and increased over the initial amount on the final day. The control and 48 h treatments showed significant increases.

### 2.6. Correlation Analysis

A significant correlation was observed between the fruit’s chilling injury and weight loss, respiration rate, ethylene production rate, electrolyte leakage, MDA, and calyx browning (Figure 7). A positive correlation was further shown to exist between chilling injuries and weight loss, MDA, electrolyte leakage, and calyx browning. Although firmness was not correlated, a relationship is also speculated.

## 3. Discussion

Fruit with tropical or subtropical origins appears to be the most susceptible to chilling injury. Sweet peppers happen to fall into this category, and previous studies have shown short-term high CO_2_ treatment to alleviate postharvest disorder due to its effect on redox and the fruit’s cellular membrane [10]. Chilling injuries in low-temperature storage include one of the disorders it alleviates [10,13]. This study compares the effects of pretreatment with 10% CO_2_ for 12, 24, and 48 h on chilling injury alongside those of untreated and modified atmosphere-packed (MAP) fruits with 50,000 cc·m^−2^·day^−1^·atm^−1^ OTR film during cold storage.

Following pretreatments, MAP, and the storage of fruits, the control fruits lost the most initial weight, while the MAP fruit lost the least. This higher weight loss in control fruits than in other treatments is believed to be caused by the fruit respiration rate (Table 1), while the lowest weight loss shown for MAP fruit is due to the MAP’s additional ability to retain moisture [5]. One of the major advantages of using CO_2_ is its ability to reduce fruits’ respiration rates [8]. This high respiration rate in the control group results in more water loss than other treatments. Keeping sweet pepper water loss under control is critical because it is inextricably linked to a number of other factors. Previous research has shown this to have a serious impact on quality, shelf life, and chilling injuries [1,16].

Fruit water loss causes the firmness to deteriorate. A significantly reduced water loss rate was observed in the MAP fruits due to water loss retentivity [5], which resulted in the highest firmness shown. Although determining this during the fruit’s storage was difficult, it was assumed to follow the trend before unpacking. This result may show that there was more cell wall degradation in the control group. Due to its ability to reduce firmness loss, CO_2_ treatment has also gained traction in postharvest management. This happens because it allows acetaldehyde to react with soluble tannins to render them insoluble, thereby increasing the fruit’s firmness [14]. In some other studies, a loss of firmness in fruit is considered a sign of chilling injury [17].

In this study, the observed differences in chilling injury were thought to be primarily due to the differences in the fruit weight loss rates, which is supported by studies that show how water loss affects the chilling injury of sweet peppers [1,16]. Fruit water loss is principally associated with the fruit’s respiration rate. This is because respiration is a metabolism process that involves the breakdown of carbohydrates, which ends up producing water as its end product. This increased respiration resulted in more water loss in the control fruit, while the MAP’s ability to retain water favored the MAP fruits. Water loss has been shown to involve some cellular disintegration and membrane integrity damage occurring through the fruit’s skin [1,16]. 

Calyx is not just an important factor for consumers in fruit consideration; it also protects the abscission zone from fungal attacks, increasing the fruit’s storability [18]. The browning of the calyx is found to be caused by the increased promotion of the polygalacturonic acid enzyme and cellulose triggered by ethylene accumulation [18]. However, the reduced weight loss may have also contributed to the lower browning observed for MAP fruit, as the calyx is an important point for water loss [16]. This is a physical symptom of a chilling injury, and our findings show that CO_2_ treatment reduces it. 

Generally, a coincident relationship exists between respiration and ethylene production at harvest, which is usually due to the increased endogenous ethylene level. However, several internal and external factors also affect the respiratory and ethylene metabolisms. The increase in respiration rate observed after CO_2_ treatment could probably be due to induction, which is sometimes accompanied by the ATP content [19]. Although further speculation was made, if the increase in respiration results from a treatment higher than 10% CO_2_, which may be due to injury depending on the tissue type, cultivar, and duration of exposure [19], there is no difference in ethylene production rates because ethylene production is dependent on exposure duration, concentration, and fruit [19]. Comparing our results with another study on persimmon, similar observations were reported after treatments on respiration and ethylene production rates [10]. When fruits were moved to room temperature, the 24-h-treated fruit had the lowest respiration rate, while it was highest in MAP fruits, probably also due to induction. Throughout the room-temperature days no significant difference was observed, which was similarly reported for persimmons in the same study [10].

The ethylene production rates were not really significant, except for the lower rate observed in the 48-h-treated fruits. These results agree with what Wang [15] reported, i.e., that ethylene inhibitions stop immediately when CO_2_ treatment ends for sweet peppers. In the same study, the production rate was also shown to coincide with the calyx injury projected to be caused by high CO_2_. Even though control groups were untreated, similar observations were found in this study. Furthermore, ethylene is known as a response to cold damage; the high levels of ethylene in the control group are also evidence of chilling injury [16].

As a way of regulating the CO_2_ content in the film, the oxygen transmission film (OTR) allows the transmission of O_2_ within the fruit’s environment [11] and the fruit’s respiration is responsible for changes in the gas content. Following discussion from previous studies, peppers have a tolerance level of 5% CO_2_ in MAP conditions [5]. The 50,000 cc·m^−2^·day^−1^·atm^−1^ OTR film used maintained the average value below the tolerance level until opening time.

Contrary to what would have been expected, no difference in MDA content was seen after CO_2_ treatment. This may be because it may take time for the accumulation of undesirable respiration to take place before showing up as MDA. MDA is reported as an abnormal compound that results from anaerobic respiration or oxidative damage that accumulates in the fruit tissue [17]. This result was the same on the final day due to an insignificant difference in the fruit respiration rate (Table 1). However, it was the lowest in the 24 h treatment. This could be as a result of the CO_2_ inhibiting the succinic oxidase complex and succinic dehydrogenase, thereby slowing down respiration [9]. This finding shows that there is a link between MDA and respiration rates, as previously shown [17]. However, another study found a significant difference in MDA due to CO_2_ treatment [20].

The initial electrolyte leakage of 40.8 ± 7.56% was similar to what Lim [21] reported. The electrolyte leakage increased due to cellular stress after CO_2_ treatment. The primary aim of measuring the electrolyte leakage is to determine the extent of cellular damage that occurred due to the impact of storage conditions, treatment conditions, etc., on the fruit. A non-significant difference appeared on the final day. However, it appeared lower in treated fruits with a longer duration of treatment. The higher ion leakage in the control and 12 h treatment may be due to cell damage due to the cold storage’s impact on the fruit’s cell membrane. This result was also contrary to the significant ion leakage difference reported for papaya [20].

The etiology of the changes in soluble solids after treatment is not fully understood, as peppers’ soluble solids sometimes do not follow the expected pattern [16]. However, the increase in the control group on the last day may have been caused by senescence. The ethylene production of fruit was found to increase significantly more than in other treatments (Table 2), while the cause of the increase in the 48 h treatment could be a defense mechanism in response to chilling injury [17]. 

A significant correlation was observed between the fruit’s chilling injury and weight loss, respiration rate, ethylene production rate, electrolyte leakage, MDA, and calyx browning. Therefore, these parameters can be used as a reliable tool to determine the impact of chilling injuries on sweet pepper fruit. A positive correlation was further shown to exist between chilling injury and weight loss, MDA, electrolyte leakage, and calyx browning, which could better explain why chilling injury increases over time during storage. This is not surprising given the established correlation between chilling injuries and these parameters [1].

## 4. Materials and Methods

### 4.1. Fruit Treatment and Storage Conditions

Mature sweet pepper Nagno (Rijk Zwaan) cultivars were harvested from a greenhouse at Kangwon National University and transported to the laboratory for the experiment. Before all else, the fruits were sorted and evenly divided into groups. The fruits were pre-treated with 10% CO_2_ in an airtight container for 12, 24, and 48 h, respectively, before cold storage. In addition, throughout the cold storage, some were also packed in MAP with a 50,000 cc·m^−2^·day^−1^·atm^−1^ oxygen transmission rate (OTR) film. This film was selected following its outcome in the preliminary experiment, as it regulates CO_2_ within pepper tolerance [5]. Prior to and after the CO_2_ treatment, the respiration and ethylene production rates, brix, electrolyte leakage, and malondialdehyde content were measured to determine the impact of the CO_2_ treatment on these parameters. Untreated fruits were stored at 5 °C until the end of treatment, while treatment was also carried out at 5 °C. The end of treatment was considered the start of the experiment, with the control and MAP groups stored according to their categories. The fruits were stored for 15 days at 5 °C with a RH of 83%, and for another 8 days at room temperature. 

### 4.2. Malondialdehyde (MDA) Assay and Electrolyte Leakage

The malondialdehyde was measured using the thiobarbituric-acid-reactive substance (TBARS) in reference to Wang [22]. The thiobarbituric acid and trichloroacetic acid chemicals were bought from Daejung and Alfa Aesar, respectively. In total, 20 mL of 10% trichloroacetic acid was used to homogenize the 4.0 g of pulp tissue from 10 pepper fruits, which were subsequently centrifuged (Hanil Mega 17R Centrifuge) at 5000× *g* for 10 min. Then, 3 mL of 0.5% thiobarbituric acid that had previously been dissolved in 10% trichloroacetic acid was added to 1 mL of the supernatant. To clarify the precipitation, the reaction mixture solution was heated for 20 min at 95 °C, quickly cooled with ice, and centrifuged for 10 min at 10,000× *g*. The non-specific absorbance at 600 nm was measured with a spectrophotometer (BioMate 3S UV-Vis), and the absorbance at 532 nm was removed. The amount of MDA was calculated as the μM kg^−1^ of fresh weight (FW) using an attenuation value of 155 mM^−1^ cm^−1^.

Electrolyte leakage (EL) was determined following the Wang [11] method. Sweet pepper mesocarp samples (0.5 g) were immersed in 0.4 M mannitol (25 mL) and shaken for 3 h at room temperature, after which the EL was measured immediately with a handheld meter (HI 9813-6 Portable pH/EC/TDS/°C Meter; Hanna Instruments, Padova, Italy). Then, the samples were frozen at −20 °C and thawed at ambient temperature; this was repeated, and the solution was measured twice. The EL was calculated as follows:(1)$$\mathrm{Relative}\;\mathrm{electrolyte}\;\mathrm{leakage}\;(\mathrm{EL})=\frac{EL_{i}}{EL_{f}}\times 100\%$$

$EL_{i}$ means initial electrolyte leakage, while $EL_{f}$ means final electrolyte leakage.

### 4.3. Gas Production Rates and Concentration in Modified Atmosphere Packaging (MAP)

The ethylene production and respiration rates were measured after leaving the fruits at room temperature in an airtight container (1050 mL) for no less than 3 h. The GC-2010 Shimadzu gas chromatograph (GC-2010, Shimadzu Corporation, Tokyo, Japan) and infrared CO_2_ analyzer (Model Check Mate 9900, PBI-Dansensor, Ringsted, Denmark) were used to measure the ethylene production and respiration rate, respectively. Moreover, 1.0 mL of gas was collected from the headspace of airtight containers and MAP, which was passed into the GC, while the respiration rate was determined by similarly measuring the CO_2_ in the headspace with an infrared CO_2_ analyzer to determine the respiration rate. The GC-2010 conditions were such that it was equipped with a BP 20 Wax column (30 m × 0.25 mm × 0.25 µm, SGE Analytical Science, Australia) and a flame ionization detector. The gas detector was set to run at 200 °C, while the oven was set at 50 ℃ and the gas carrier flow rate was 1.76 mL^−1^ min. 

### 4.4. Weight Loss and Firmness 

The weight loss rate is expressed in percentage using the formula below [16,23]:(2)$$\mathrm{Fresh}\;\mathrm{weight}\;\mathrm{loss}\;\mathrm{rate}\;(\%)=\frac{Initial\;fresh\;weight-Final\;fresh\;weight}{Initial\;fresh\;weight}\times 100\%$$

The firmness (N) was measured with a rheometer (Compac-100, Sun Scientific Co., Ltd., Tokyo, Japan) using a probe (Ø 8.0 mm) at 1.0 mm/s speed and a distance of 15 mm.

### 4.5. Chilling Injury Index and Calyx Browning 

The chilling injury index was determined in reference to Zuo [23]. The fruits were rated on a 0–4 point scale after careful examination by a 3-member panel. Zero points were given to fruits with no injury, one point was given for less than 5% injury, two points were given for 5 to around 25% injury, three points were given for 26 to about 50% injury, and four points were given for more than 50% injury. The formula below was then used to calculate:(3)$$\mathrm{CI}=\frac{\Sigma \;\left(CI\;scale\;\left(0-4\right)\times the\;number\;of\;corresponding\;fruit\;within\;each\;class\right)\;}{Total\;number\;of\;fruit\;estimated}$$

Calyx browning was also determined in a similar manner by the same people. A 5-point rating, as used by Afolabi [16], was employed. The corresponding scores were then expressed as an average. 

### 4.6. Soluble Solids Content

A pocket refractometer was used to determine the fruit’s soluble solid content (PAL-1, Atago, Tokyo, Japan). Fruit juice was thrust out and made to drop directly onto the refractometer sensor by gauze-wrapping the chopped fruit’s sample pieces, and the result was shown as brix.

### 4.7. Statistical Analysis 

The data were analyzed using GraphPad Prism 8.0 (GraphPad Software, San Diego, CA, USA) and Microsoft Excel 2016. The treatment was replicated four times. Tukey’s multiple-comparison test of a two-way ANOVA was used to compare the means, and the statistical significance was determined at *p* ˂ 0.05. Pearson correlation analysis was used to obtain the correlation coefficients of the parameters measured.

## 5. Conclusions

In postharvest management, pre-storage CO_2_ treatment has been shown to alleviate postharvest disorders. Its effects on sweet pepper chilling injuries were reported in this study. The fruits were pretreated with 10% CO_2_ for 12, 24, and 48 h, with some packed with 50,000 cc·m^−2^·day^−1^·atm^−1^ OTR film MAP throughout cold storage. For the chilling indicator indexes, the results show no highly significant differences, particularly among pretreated fruits. Therefore, considering the economic aspect, pre-storage CO_2_ treatment may not be efficient in reducing sweet peppers’ chilling injury. However, 24 h pre-storage treatment is recommended on a commercial basis for pre-storage consideration. Additionally, both 24- and 48 h treatments outperformed untreated samples for almost all parameters measured, but a 50,000 cc·m^−2^·day^−1^·atm^−1^ OTR film MAP is considered the best treatment. The etiology of CO_2_ treatment is a little complex when it comes to sweet peppers, as some factors were believed to be apparently triggered by the CO_2_ treatment.

## Figures and Tables

**Figure 1 plants-12-00671-f001:**
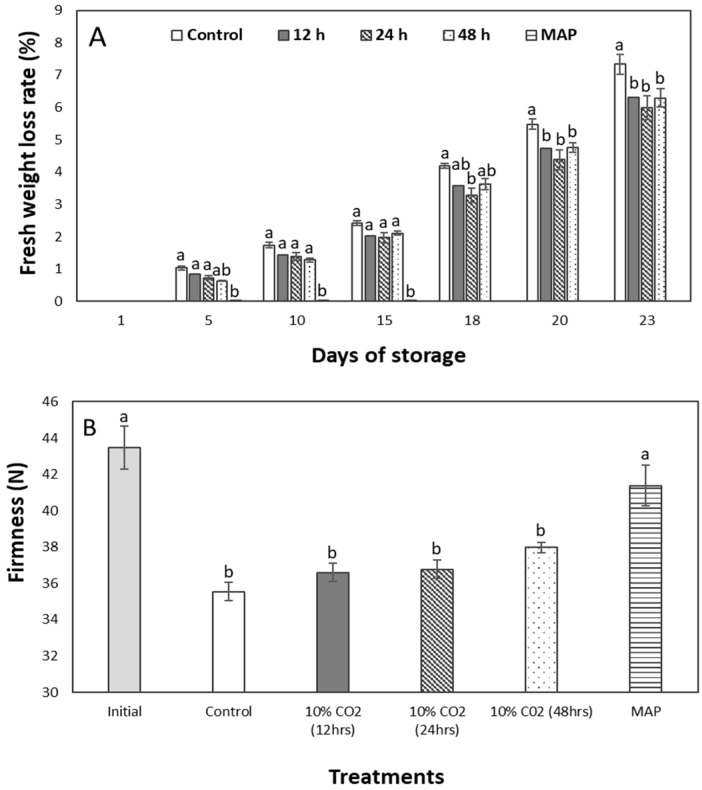
Changes in weight loss (**A**) and firmness (**B**) of sweet peppers pretreated with 10% CO_2_ for 12, 24, and 48 h, as well as untreated and modified atmosphere packaging (MAP) with 50,000 cc·m^−2^·day^−1^·atm^−1^ OTR film throughout the 15 days of cold storage and 8 days at room temperature. The vertical bars represent ± SE of *n* = 4, and different letters indicate a significant difference at *p* ˂ 0.05.

**Figure 2 plants-12-00671-f002:**
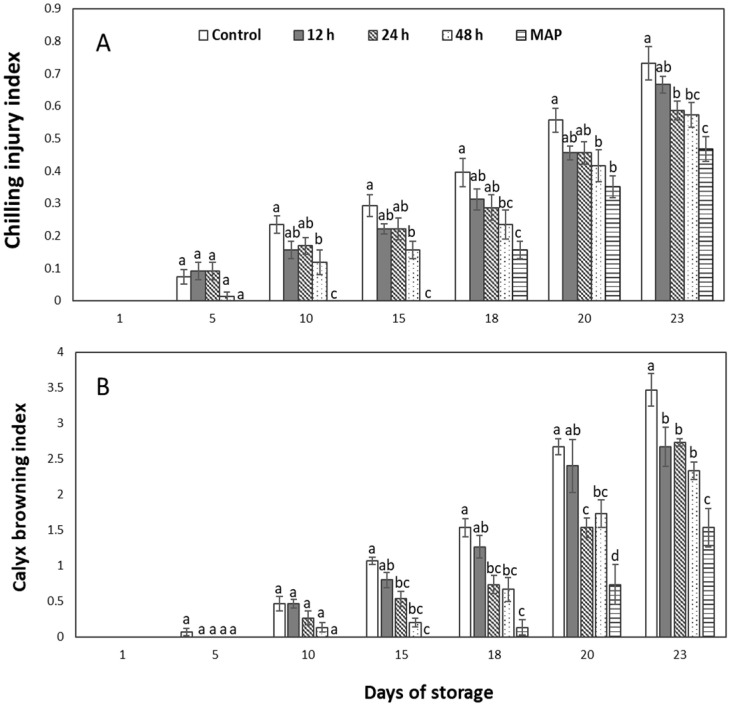
Chilling injury (**A**) and calyx browning index (**B**) of sweet peppers pretreated with 10% CO_2_ for 12, 24, and 48 h, as well as untreated and modified atmosphere packaging (MAP) with 50,000 cc·m^−2^·day^−1^·atm^−1^ OTR film throughout the 15 days of cold storage and 8 days at room temperature. The vertical bars represent ± SE of *n* = 4, and different letters indicate a significant difference at *p* ˂ 0.05.

**Figure 3 plants-12-00671-f003:**
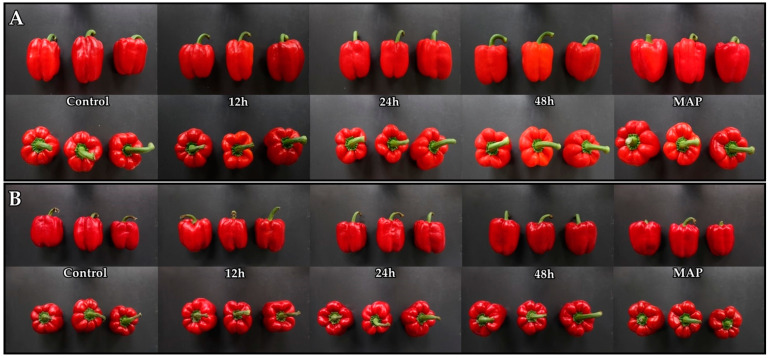
Image of sweet pepper fruits pretreated with 10% CO_2_ for 12, 24, and 48 h, as well as untreated and modified atmosphere packaging (MAP) with 50,000 cc·m^−2^·day^−1^·atm^−1^ OTR film on the 15th day of cold storage (**A**) and on the last day (**B**).

**Figure 4 plants-12-00671-f004:**
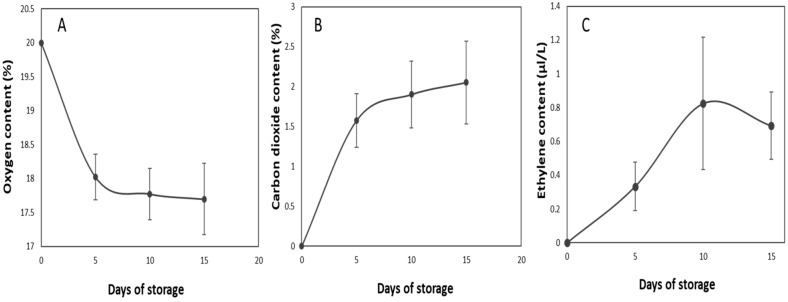
Changes in oxygen (**A**), carbon dioxide (**B**), and ethylene (**C**) content in modified atmosphere packaging (MAP) with 50,000 cc·m^−2^·day^−1^·atm^−1^ OTR film MAP throughout cold storage. The vertical bars represent ± SE of *n* = 4.

**Figure 5 plants-12-00671-f005:**
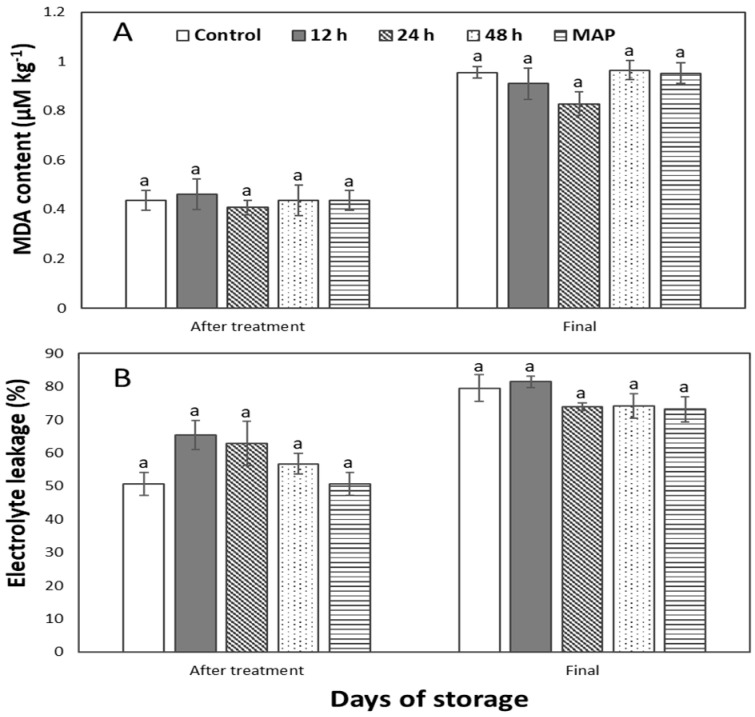
Malondialdehyde (MDA) content (**A**) and electrolyte leakage (**B**) of sweet peppers pretreated with 10% CO_2_ for 12, 24, and 48 h, as well as untreated and modified atmosphere packaging (MAP) with 50,000 cc·m^−2^·day^−1^·atm^−1^ OTR film after treatments and the last day of storage. The vertical bars represent ± SE of *n* = 4, and different letters indicate a significant difference at *p* ˂ 0.05.

**Figure 6 plants-12-00671-f006:**
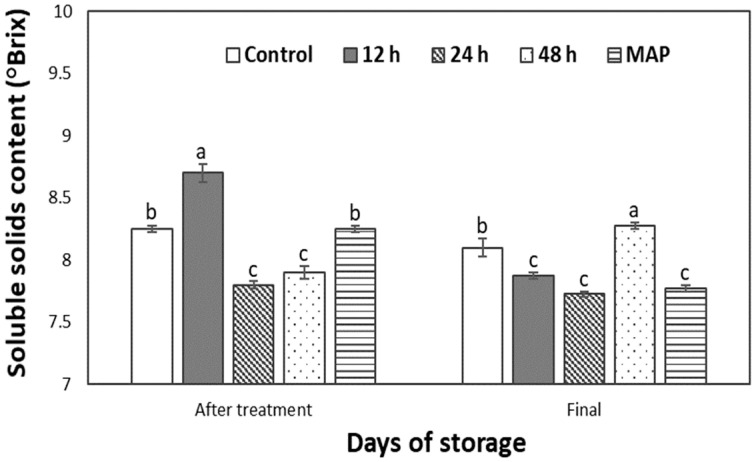
Soluble solids content of sweet peppers pretreated with 10% CO_2_ for 12, 24, and 48 h, as well as untreated and modified atmosphere packaging (MAP) with 50,000 cc·m^−2^·day^−1^·atm^−1^ OTR film after treatment and the last day of storage. The vertical bars represent ± SE of n = 4 and different letters indicate a significant difference at *p* ˂ 0.05.

**Figure 7 plants-12-00671-f007:**
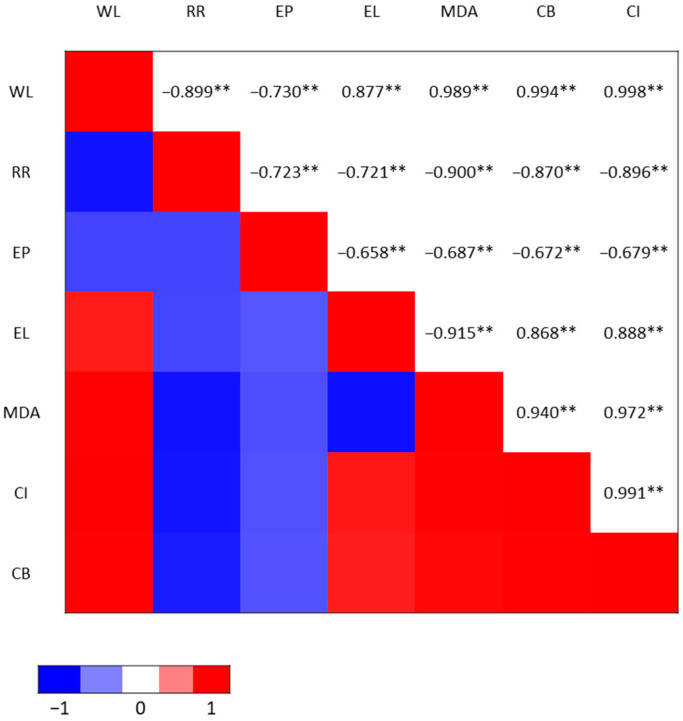
Pearson correlation coefficient heat map of weight loss (WL), respiration rate (RR), ethylene production rate (EP), electrolyte leakage (EL), malondialdehyde (MDA), calyx browning index (CB), and chilling injury index (CI) of fruits pretreated with 10% CO_2_ for 12, 24, and 48 h, as well as untreated and modified atmosphere packaging (MAP) with 50,000 cc·m^−2^·day^−1^·atm^−1^ OTR film throughout the 15 days of cold storage and 8 days at room temperature. ** indicate correlation is significant at *p* ˂ 0.01.

**Table 1 plants-12-00671-t001:** Changes in respiration and ethylene production rates of sweet peppers pretreated with 10% CO_2_ for 12, 24, and 48 h, as well as untreated and modified atmosphere packaging (MAP) with 50,000 cc·m^−2^·day^−1^·atm^−1^ OTR film throughout the 15 days of cold storage and 5 days at room temperature.

	Respiration Rate (CO_2_ mg kg^−1^ h^−1^)					
Treatment	Harvest	After Treatment	Day 15	Day 18	Day 20	Treatment
Control	12.50 ± 0.67	13.19 ± 1.15 b ^z^	16.77 ± 1.06 ab	5.22 ± 0.36 a	7.10 ± 0.28 a	**
12 h		13.82 ± 0.50 b	16.73 ± 0.96 a	6.90 ± 1.06 a	7.30 ± 0.53 a	***
24 h		18.77 ± 0.73 a	13.37 ± 0.75 b	5.54 ± 0.41 a	6.90 ± 0.45 a	***
48 h		19.91 ± 1.78 a	17.15 ± 0.90 ab	5.72 ± 0.15 a	6.25 ± 0.58 a	***
MAP		13.19 ± 1.15 b	18.04 ± 0.93 a	5.76 ± 0.23 a	7.45 ± 0.22 a	***

^z^ Values are represented as the mean ± SE (*n* = 4), and different alphabets within columns represent significant differences among treatments for each measuring day using Tukey’s multiple comparison test. Day 18 and Day 20 represent three and five days after cold storage, respectively. **, ***: significant at *p* ≤ 0.01 and 0.001.

**Table 2 plants-12-00671-t002:** Changes in ethylene production rates of sweet peppers pretreated with 10% CO_2_ for 12, 24, and 48 h, as well as untreated and modified atmosphere packaging (MAP) with 50,000 cc·m^−2^·day^−1^·atm^−1^ OTR film throughout the 15 days of cold storage and 5 days at room temperature.

	Ethylene Production Rate (µL kg^−1^ h^−1^)					
Treatment	Harvest	After Treatment	Day 15	Day 18	Day 20	Treatment
Control	2.10 ± 0.07	2.45 ± 0.40 a ^z^	1.14 ± 0.08 ab	1.16 ± 0.17 a	1.29 ± 0.08 a	*
12 h		1.91 ± 0.70 a	1.63 ± 0.05 a	0.94 ± 0.04 a	0.77 ± 0.09 ab	*
24 h		1.70 ± 0.65 a	1.30 ± 0.18 ab	0.99 ± 0.09 a	0.81 ± 0.09 ab	*
48 h		2.52 ± 0.87 a	0.78 ± 0.04 b	1.07 ± 0.09 a	0.57 ± 0.02 b	***
MAP		2.45 ± 0.40 a	1.04 ± 0.12 ab	1.19 ± 0.13 a	0.67 ± 0.07 ab	*

^z^ Values are represented as the mean ± SE (*n* = 4), and different alphabets within columns represent significant differences among treatments for each measuring day using Tukey’s multiple comparison test. Day 18 and Day 20 represent three and five days after cold storage, respectively. *, ***: significant at *p* ≤ 0.5 and 0.001.

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
