# Peer review of "Effect of Pre-Storage CO2 Treatment and Modified Atmosphere Packaging on Sweet Pepper Chilling Injury"

_plants, 2023, doi:10.3390/plants12030671_

Round 1

Reviewer 1 Report

The study was aimed to investigate the effect of pre-storage CO2 treatment and modified atmosphere packaging on Sweet Pepper chilling injury. Currently, there are numerous manuscripts on this topic. Apart from that, additional analysis and/or full explanation of the results that should be necessary to improve the manuscript.

1.      The author should provide the morphology changes during storage.

2.      The format of CO2 and O2 in figures need revision.

3.      The author should provide the physiological characteristics (H2O2, MDA, Soluble solids content …) during storage.

4.      Additional analysis and/or full explanation of the results are needed. This would explain the results and improve the manuscript.

5.      Line 180-181:” Chilling injuries in low-temperature storage are one of the reported disorders it alleviates.” Please rephrase this part.

6.      The discussion should be somewhat modified by giving more details on the difference between this study and many previous works, especially their new discoveries.

7.      References format in the text need revision.

Author Response

The authors sincerely appreciate the editor’s and reviewers’ comments for the first round of revision. We’ve responded to all the reviewers’ comments and modified the manuscript accordingly. Please let us know if you have any further questions or comments. We appreciate your assistance in revising our manuscript, and we find your comments helpful.

Responses to questions, suggestions, and new comments in the manuscript are written in red.

Review of plants-2153562

The authors have thoroughly and thoughtfully addressed the reviewers’ comments and responses to questions and suggestions are as follow:

1. The author should provide the morphology changes during storage.

Reply: We respond to this comment by including the fruit’s phenotype during the experiment, which can be found in line 102.

Figure 3. The phenotype image of sweet pepper fruits pretreated with 10% CO2 for 12, 24, and 48 hours, as well as untreated and modified atmosphere packaging (MAP) with 50,000 cc·m–2·day–1·atm–1 OTR film on the 15th day of cold storage (A) and on the last day (B).

2. The format of CO2and Oin figures needs revision.

Reply: The correction to this has been made and can be found in line 135 of the manuscript.

3. The author should provide the physiological characteristics (H2O2, MDA, Soluble solids content …) during storage.

Reply: The physiological characteristics weren’t measured during storage because no highly significant differences were seen in the factors that may warrant the changes in those parameters. We believe the provided phenotypic image would support this.

4. Additional analysis and/or full explanation of the results are needed. This would explain the results and improve the manuscript.

Reply: We respond to this by saying we think additional analysis may not be needed since there weren’t big differences for the major chilling indicators. However, we have explained our results further, as suggested.

5. Line 180-181:” Chilling injuries in low-temperature storage are one of the reported disorders it alleviates.” Please rephrase this part.

Reply:  This sentence has been rephrased and can be found in lines 194-195 in the manuscript. Chilling injuries in low-temperature storage include one of the disorders it alleviates.

6. The discussion should be somewhat modified by giving more details on the difference between this study and many previous works, especially their new discoveries

Reply: We have modified the discussion and compared our results with other results from similar studies. For instance, lines 272-273 “However, another study found a significant difference in MDA due to CO2 treatment [20],” and lines 281-282 “This result was also contrary to the significant ion leakage difference reported for papaya,’’ which are very crucial in this study.

7. References format in the text need revision

Reply: We respond to this by saying that the references have been adjusted.

We are thankful for your constructive comments!

Reviewer 2 Report

Plants-2153562

The work reported in this manuscript is a careful study of the Effect of Pre-storage CO2 Treatment and Modified Atmosphere Packaging on Sweet Pepper Chilling Injury and the research group has been studying the storage of sweet pepper for some time.

A large number of properties related to chilling injury of the sweet pepper were measured and correlated, allowing identifying the best treatment. The discussion of the results is supported by previous studies in the field listed in the references. The number of references is not very large and a third of them date from the 20th century. Reviewing the list is mandatory as references 1-15 appear twice.

A first comment on the manuscript sequence. This comment is mainly for the editor of the journal. In the Template for Plants “Materials and methods” is the 4th section, after results and discussion. Hard for me to understand the reasoning behind. How can the reader understand the figures and the results discussion before reading how properties were measured and indexes calculated? The reader needs this information sooner. At least I do!

Therefore, I will start with comments on section 4 Materials and Methods

·       a) Information on chemicals used is needed: seller, purity, …

·        b) Equipment characteristics (brand and model) are missing for: centrifuge (line 292), equipment used to measure absorbance (line 296), freezer (line 303).

·       c)  Line 289 - 2.2  (4.2?) Malondialdehyde (MDA) Assay and Electrolyte Leakage

More suggestions and questions:

·        d)  In Results, line 21- please consider writing that “control group” corresponds to untreated peppers. This information is in the figure captions, but would be more informative if included in text.

·       e) In Results, line 154, the increases reported in soluble solids content refer to before or to after treatment? If after treatment, please clarify the increase in the control group.

·      f)   In Figure 3 changes in oxygen and carbon dioxide content as well as the ethylene concentration during cold storage are shown. Could this study have been extended for the period at room temperature? What results would you expect then? The data in Tables 1 and 2 are shown in Figure 3?

·      g)   In Figure 6 please confirm the parameters at the end of the axes: CB and CI (calyx browning and chilling injury) in both or it should change to CI and CB?

Author Response

The authors sincerely appreciate the editor’s and reviewers’ comments for the first round of revision. We’ve responded to all the reviewers’ comments and questions. Please let us know if you have any further questions or comments. We appreciate your assistance in revising our manuscript, and we find your comments helpful.

Responses to questions, suggestions, and new comments in the manuscript are written in red.

Review of plants-2153562

The authors have thoroughly and thoughtfully addressed the reviewers’ comments and responses to questions and suggestions are as follow:

a. Information on chemicals used is needed: seller, purity…

Reply: The information about this has been included in the manuscript and can be found in lines 315-316. “The thiobarbituric acid and trichloroacetic acid chemicals were bought from Daejung and Alfa Aesar, respectively.”

b. Equipment characteristics (brand and model) are missing for: centrifuge (line 292), equipment used to measure absorbance (line 296), freezer (line 303).

Reply: Information about this has been provided and can be found in lines 318-319, “which were subsequently centrifuged (Hanil Mega 17R Centrifuge),” line 321 (“quickly cooled with dry ice),” and lines 322-323, “The non-specific absorbance at 600 nm was measured with a spectrophotometer (BioMate 3S UV-Vis).”

c. Line 289 - 2 (4.2?) Malondialdehyde (MDA) Assay and Electrolyte Leakage

Reply: This has been corrected now in the manuscript and can be found in line 313 now, “4.2 Malondialdehyde (MDA) Assay and Electrolyte Leakage.”

d. In Results, line 21- please consider writing that “control group” corresponds to untreated peppers. This information is in the figure captions, but would be more informative if included in text.

Reply: This has been included in the text as a bracket. The correction can be found in line 73, "in the control group (untreated fruits)."

e. In Results, line 154, the increases reported in soluble solids content refer to before or to after treatment? If after treatment, please clarify the increase in the control group.

Reply: We meant that we observed an increase when rechecked after treatments. However, the sentence has been modified in that part. Lines 161-167 “The soluble solids content was 7.43 ± 2.3 °Brix before treatment, and Figure 5 depicts the soluble solids content after CO2 treatment and on the final day. It increased when rechecked after CO2 treatment. It increased for all fruits but was at a lower rate for 24-hour and 48-hour treatments compared to others. Although it was not monitored during storage because no highly significant differences were seen in respiration or ethylene production rates (Tables 1 and 2), it varied for all fruits and increased over the initial amount on the final day. The control and the 48-hour treatments showed significant increases.”

f. In Figure 3 changes in oxygen and carbon dioxide content as well as the ethylene concentration during cold storage are shown. Could this study have been extended for the period at room temperature? What results would you expect then? The data in Tables 1 and 2 are shown in Figure 3?

Reply: The questions are intelligent ones, and we are grateful you followed them raptly. This study wasn’t extended for a period at room temperature. Also, we expect the CO2 within the packages to increase if they were extended at room temperature, as we observed in our preliminary studies, and this will result in a higher CO2 level beyond the fruit’s tolerance, which can harm the fruit. Lastly, the data in Tables 1 and 2 are not the same as the data shown in Figure 3. Figure 3 shows the gas concentrations (oxygen, carbon dioxide, and ethylene) for the modified-atmosphere packaged (MAP) fruits before they were opened. MAP is used in this study because it modifies the environment by containing the fruit’s respiration. As a result, the CO2 content of packages increases. In this study, we investigated the long-term effect of fruits in a higher-than-air CO2 environment.

g. In Figure 6 please confirm the parameters at the end of the axes: CB and CI (calyx browning and chilling injury) in both or it should change to CI and CB?

Reply: We respond to the question by saying that we feel it should not be in both because we have measured and correlated them independently.

We are thankful for all your comments!

Round 2

Reviewer 1 Report

The manuscript has been satisfactorily revised. The authors have answered to most comments in an appropriate manner.